# Modern Methods for Assessing the Quality of Bee Honey and Botanical Origin Identification

**DOI:** 10.3390/foods9081028

**Published:** 2020-07-31

**Authors:** Anna Puścion-Jakubik, Maria Halina Borawska, Katarzyna Socha

**Affiliations:** Department of Bromatology, Medical University of Bialystok, Mickiewicza 2D St., 15-222 Białystok, Poland; bromatos@umb.edu.pl (M.H.B.); katarzyna.socha@umb.edu.pl (K.S.)

**Keywords:** adulteration, bee honey, quality assessment, variety

## Abstract

This paper is a summary of the latest literature on methods for assessing quality of natural bee honey. The publication briefly characterizes methods recommended by the International Honey Commission, published in 2009, as well as newer methods published in the last 10 years. Modern methods of assessing honey quality focus mainly on analyzing markers of individual varieties and classifying them into varieties, using, among others, near infrared spectroscopy techniques (NIR), potentiometric tongue, electronic nose, nuclear magnetic resonance (NMR), zymography, polymerase chain reaction (PCR), DNA metabarcoding, and chemometric techniques including partial least squares (PLS), principal component analysis (PCA) and artificial neural networks (ANN). At the same time, effective techniques for analyzing adulteration, sugar, and water content, hydroxymethylfurfural (HMF), polyphenol content, and diastase activity are being sought. Modern techniques enable the results of honey quality testing to be obtained in a shorter time, using the principles of green chemistry, allowing, at the same time, for high precision and accuracy of determinations. These methods are constantly modified, so that the honey that is on sale is a product of high quality. Prospects for devising methods of honey quality assessment include the development of a fast and accurate alternative to the melissopalynological method as well as quick tests to detect adulteration.

## 1. Introduction

Natural bee honey is a sweet product made by honeybees *Apis mellifera* L., both from the nectar of plants and from the excreta of insects sucking the juice from living parts of plants or from secretions of live parts of plants. Those are then combined with specific secretions of bees, stored in honeycombs, evaporated, and left in honeycombs to mature. The Council of the European Union distinguishes nectar and honeydew honey as well as filtered honey, pressed honey, extracted honey, drained honey, comb honey, and chunk honey or cut comb honey [1].

Bee honey intended for human consumption should be of sufficiently high quality. Requirements in individual countries include parameters such as variety determination, water content (no more than 20%; no more than 23% in heather and baker′s honey; no more than 25% in baker′s honey from heather), HMF (general, no more than 40 mg/kg), proline (no less than 25 mg/100 g), diastase activity (general, no less than 8 on the Schade scale), electrical conductivity (e.g., in honeydew honey, no less than 0.8 mS/cm), pH, insoluble impurities (no more than 0.1 g/100 g) and free acidity (in general, no more than 50 milli-equivalents acid/1000 g). Other parameters include, for example, color and total phenolic content (TPC). To assess honey quality, standard methods, including spectrophotometric, refractometric, titration and melissopalynological methods, are used [2].

Growing scientific interest in natural bee honey is demonstrated by an increasing number of publications available in the PubMed article database (Figure 1).

Bee honey has been used for centuries as a nutritional, preventive, and medicinal product. Over the past 10 year, its use has been studied in the following diseases: arthritis [3], breast cancer (cell line) [4], cervical cancer (cell line) [4], colon cancer (cell line) [5], cough [6], type 2 diabetes [7], hepatic steatosis [8], influenza [9], *glioblastoma multiforme* (cell line) [10], *Helicobacter pylori* [11], mucositis [12], osteoporosis [13], pressure ulcers [14], *Pseudomonas* [15], renal cancer cell line [16], and rosacea [17]. Due to such diverse activity, honey must be of the highest quality [1,2,18,19].

Market requirements are very stringent: results should be obtained as soon as possible; the method should be characterized by the highest accuracy and precision; and, in addition, it should be inexpensive. Therefore, many authors publish papers on new quality analysis methods. The aim of the present study was to summarize the most important methods of assessing honey quality published in the last 10 years. The novelty of these publication lies in the collection of data regarding both the evaluation of honey composition as well as adulteration.

## 2. Honey Composition

Natural bee honey is a complex product containing several hundred compounds belonging to various chemical groups. In addition to such basic groups as those listed in Table 1, i.e., glucose, fructose, sucrose, water, organic acids, minerals, or amino acids, bee honey contains, among others, polyphenolic compounds, dyes, vitamins, essential oils, and other active substances.

Twenty five different sugars are found in honey. In nectar honeys, the sugars include erlose, maltose, sucrose, and turanose, while in honeydew honey-melezitose and raffinose. Dextrins are found in Italian metcalfa honey. Among the acids in honey, there is mainly gluconic acid, and also in smaller amounts: acetic, citric, formic, lactic, maleic, malic, oxalic, pyroglutamic, and succinic acids. These acids influence the pH of honey, which is commonly between 3.3 and 4.6. Among the main minerals in honey, potassium should be mentioned (it constitutes approximately 33% of all elements present in honey). Others include barium (from 0.01 to 0.08 mg/100 g), boron (from 0.05 to 0.3 mg/100 g), chlorine (from 0.4 to 56 mg/100 g), cobalt (from 0.1 to 0.35 mg/100 g), fluoride (from 0.4 to 1.34 mg/100 g), iodine (from 10 to 100 mg/100 g), lithium (from 0.225 to 1.56 mg/100 g), molybdenum (up to 0.004 mg/100 g), nickel (up to 0.051 mg/100 g), rubidium (from 0.04 to 3.5 mg/100 g), silicium (from 0.05 to 24 mg/100 g), strontium (from 0.04 to 0.35 mg/100 g), sulfur (from 0.7 to 26 mg/100 g), vanadium (up to 0.013 mg/100 g), and zirconium (from 0.05 to 0.08 mg/100 g). The elemental composition is influenced by the origin-honeydew honeys are characterized by a higher content of elements. The most important and abundant amino acid in honey is proline. Moreover, enzymes such as amylase (diastase), invertase, catalase, and glucose oxidase are present in honey. Enzymatic content proves the freshness of honey, and proper heating and storage conditions. Volatile compounds, both from the nectar and bee secretions, determine the smell of honey. Over 600 of them have been identified to date. Dark honeys contain more flavonoids. When honey is stored at too high a temperature and in the process of prolonged heating (particularly honeys with a lower pH), the content of the unfavorable component, HMF, increases [20].

## 3. Methods for Assessing Bee Honey Quality

Natural bee honey has been the subject of research for several decades. However, it still surprises scientists with its diversity, nutritional, prophylactic, and biological activity. Therefore, techniques that enable the assessment of its quality are continually modified and improved with the aim of reducing analysis time, eliminating expensive and harmful reagents, decreasing workload, and increasing accuracy.

### 3.1. Variety of Honey

The melissopalynological method is the principal technique used to determine the variety of bee honey. This technique enables determination of the share of predominant pollen grains in a particular honey, on the basis of which the honey variety is named, with the name derived from the botanical name of the plant or plants. The method also allows for identification of adulterated honeys [21].

The melissopalynological method was developed in 1895 by Pfister. It has been improved since then and is currently the most widely used technique for correct determination of honey variety. Beekeepers, however, determine the variety of honey on the basis of its organoleptic characteristics and observations from which plants bees collect nectar [22]. Table 2 shows modern methods for identifying the type and variety of honey. They are briefly characterized in a further part of the paper.

Determination of specific optical rotation is a method that distinguishes nectar honeys (usually with negative values) from honeydew honeys (with positive values). Specific optical rotation [α] $\frac{D}{20}$ is the angle of rotation of polarized light at the wavelength of the D line of sodium, at 20 °C, of an aqueous solution containing 1 of the substance. The measurement is performed using the polarimetric method and differences between honey types are due to their carbohydrate composition [2].

A number of researchers are seeking techniques that will not be as difficult and time-consuming to perform as the melissopalynological method. One of the newest approaches to identifying honey varieties uses an electronic tongue. An electronic tongue is a modern device which can be used to distinguish between honey samples in conjunction with methods of optical spectroscopy, Ultraviolet-Visable-Near Infrared (UV-VIS-NIR), and statistical analysis: cluster analysis (CA) and principal component analysis (PCA). By combining these techniques and using with multidirectional principal component analysis, correct classification of samples was obtained in 100% of cases [23].

Differences between the profiles of volatile compounds for 49 linden honeys (*Robinia pseudoacacia* L.) and 16 chestnut honeys (*Castanea sativa* Mill.) provided the basis for the development of an electronic tongue. It is a matrix with 22 sensors and pattern recognition software which aims to imitate the human sense of taste to classify products. In a study by Čačić et al. [24], honey samples were incubated at room temperature for 20 min, then the following phases were programmed: standby (20 °C, 10 min) and incubation (40 °C, 5 min). Measurement time was 5 min. PCA was used to show differences between honey samples from different geographical regions. The five main components explained 66.26% of the variability. The analysis revealed similarities in samples from geographic regions that were in close proximity to each other; therefore, it can be a tool for determining the geographical origin of honey [24].

Scientists from Portugal used an electronic potentiometric tongue to identify honeys from the following plants: *Castanea* sp., *Echium* sp., *Erica* sp., *Lavandula* sp., *Prunus* sp., *Rubus* sp., yielding 100% classification compliance [25].

Dymerski et al. [26] published a paper on the classification of bee honeys from Poland using an electronic nose. The authors based their research on the use of six semiconductor sensors and a pneumatic system which caused honey samples to bubble, which together formed the electronic nose. This technique makes it possible to distinguish, based on volatile compounds, between the following varieties of honey: acacia, buckwheat, honeydew, linden, and rapeseed.

The nuclear magnetic resonance (NMR) technique has been used to distinguish honeydew and nectar honeys obtained from oak (*Quercus* L.) [27]. The determinant of the botanical origin of this variety is quercitol. A considerable advantage of this method is speed of analysis.

The subject of research by Rossano et al. [28] was the creation of a specific fingerprint by two-dimensional zymography. Proteins present in honey can come from either nectar, pollen grains, or gland secretions of the honeybee. The most important proteins found in honey are nine types of Major royal jelly protein-1 (MRJP-1). The authors studied the proteolytic properties of varietal honeys (orange, chestnut, eucalyptus, and sulla, which is obtained from *Hedysarium coronarium* L.) and their protein profile. Eucalyptus honeys were characterized by the highest protein content (0.91–1.24 mg protein/g honey), while chestnut honeys—by the highest total proteolytic activity (25.29–36.43 mU/mg protein). By way of illustration, for orange honeys, 3 groups were obtained in the image (first—19 kDa, isoelectric point 8.5–9.3; second—20 kDa, isoelectric point 4.2–4.4; and third—24 kDa, isoelectric point 8.2–8.8). Two-dimensional zymograms can therefore provide the basis for fast classification of natural varieties of bee honey and may also enable determination of the origin of honey.

The MRJP2 gene has also been used to discriminate between two types of honey-honey produced by *Apis mellifera* and that produced by *Apis cerana*. The latter is far more expensive, and therefore there have been cases of mislabeling honey in terms of origin. In order to identify them correctly, the authors designed two pairs of species-specific primers. The amplification products of *A. mellifera* and *A. carena* honeys were 560 and 212 base pair (bp), respectively. The obtained primers were characterized by high species specificity. The MRJP2 gene was detected using the PCR method and selected primers. Differences in the gene formed the basis for establishing the origin of honey. The PCR method enabled detection of the addition of *A. mellifera* honey which was as low as 1% [29].

Attempts have been made to distinguish between bee honey varieties using the CIE scale (Commission Internationale de l’Eclairage) L * C * abh ab [30]. The authors Tuberoso et al. [30] assessed the color of 17 monofloral honeys (*n* = 305). The advantage of the proposed method is small sample size (approximately 2 g) and absence of any destructive effect (samples can be reused in subsequent analyses). PCA and hierarchical clustering analysis (HCA) have been used to distinguish between different honey varieties and classifications. In the case of reflection methods, the L * value for light honeys is below 50 while for dark honeys, it is above 50. The authors emphasize the fact that this technique may be particularly useful in the case of honeys with the protected designation of origin (PDO) mark. This concept covers honeys produced, processed, and prepared in one area which have distinct characteristics from this area. Their names are legally protected and listed on the EU protected food name register [1]. The development of rapid tests which would assist in honey identification is particularly important since bee honey is one of the 25 products at highest risk of adulteration [36,37].

Another method of honey classification is fluorescence spectroscopy and statistical analyses based on parallel factor analysis (PARAFAC) and partial least squares method combined with discriminant analysis (DAPLS) [31]. It was used in a study by Lenhardt et al. [31], which investigated acacia, sunflower, lime, meadows, and artificial honey (*n* = 95). Fluorescence in the range of excitation (260–290 nm) and emission (330–360 nm) comes from aromatic amino acids and is particularly important in differentiating the botanical origin of honey samples. The authors found that phenolic compounds and Maillard reaction products had the greatest impact on the distinction of individual varieties—emissions of these compounds differed most between honey varieties. Classification efficiency was approximately 90%. Additionally, artificial honey was detected 100% correctly.

A method of identifying botanical species of plants in multifloral honey on the basis of DNA was described by Bruni et al. [32]. The authors used rbcL and trn—psbA plastids as markers. The aim of the study was to create a database of “bar codes” of selected plants and, on that basis, to identify plants from which nectar was obtained. The application of this method allowed for, among others, the identification of nectar from the genera *Fagus*, *Quercus*, *Castanea* and from toxic plants such as *Atropa belladonna* (Wolfcream). Therefore, this method can be used indirectly to identify varieties of bee honey.

In the same year, researchers from the USA [33] published a method of analyzing plant pollen, based on the technique of DNA metabarcoding (testing of the so-called DNA bar codes). This technique is based on the analysis of DNA barcodes, i.e., short sequences. Primers specific to the second internal transcribed spacer were used in the experiment. Then, DNA amplicon, which was isolated from pollen samples collected by bees, was sequenced. The ribosomal sequences served as a specific code for the genetic identification of plant pollen. The proposed technique is particularly useful in the recognition of pollen, present in bee honeys, which is difficult to identify and rare.

The study investigated 71 acacia honeys from China, from 6 different sources. The authors analyzed 31 different factors using the partial least squares-discriminant analysis (PLS-DA) technique: 18 polyphenols, 9 oligosaccharides, and 4 carbon isotope ratio values. Based on those parameters, correct classification regarding the geographical origin of honey was obtained in 94.12% cases [34].

Researchers from New Zealand modified the technique of detecting sugar adulteration in two of New Zealand’s most famous honeys—manuka honey obtained from *Leptospermum scoparium* and kanuka honey from *Kunzea ericoides* [35]. Adding sugar, sugar cane, or High-fructose corn syrup (HFCS) to honey is a common technique used by manufacturers in order to generate higher profits. Based on the recommendations of the International Organization of Chemical Food Analysts (AOAC, Association of Official Agricultural Chemists), the authors improved the method, using the technique of continuous flow mass spectrometry by testing the ratio of isotopes to determine d13C. However, this method requires further refinement as some of the tested samples were incorrectly classified. It was mainly samples which contained less than 75% of the predominant pollen that were affected.

Many authors are looking for botanical and geographical markers—compounds characteristic of individual varieties of honey. These compounds allow for a specific variety to be clearly defined. By way of illustration, markers for 17 varietal honeys have been identified using liquid chromatography with diode array detection (LC-DAD) and/or gas chromatography mass spectrometry (GC-MS) methods. The honeys came from the following countries: Croatia, France, Germany, Hungary, Italy, Poland, Romania, Ukraine, and Spain. Selected ones are shown in Table 3. Confirmation of the presence of specific markers is a starting point for further analysis and statistical inference.

### 3.2. Sugars

The carbohydrate composition of natural bee honey may be one of the key factors in establishing its botanical origin and, indirectly, enabling its proper classification [40]. Sugars in honey are produced by enzymatic sucrose hydrolysis and transglycosylation [41].

The content of fructose, glucose, sucrose, as well as sugars such as maltose, turanose, erlose, raffinose, melezitose, and isomaltose is determined by high performance liquid chromatography (HPLC) with infrared (IR) detection. Peaks are identified based on the standards used, their retention times and peak heights. The column and detector should be thermostated at 30 °C. Results are presented in g/100 g [2]. Total glucose and fructose content should be no less than 60 g/100 g for nectar honeys, while for honeydew and honeydew nectar it should be no less than 45 g/100 g. Sucrose content should be no more than 5 g/100g, except for acacia (*Robinia pseudoacacia* L.) and alfalfa (*Medicago sativa* L.) honey (the norm: no more than 10 g/100 g), and honey obtained from lavender (*Lavandula* spp. L.) and borage (*Borago officinalis* L.) (the norm: no more than 15 g/100 g) [1]. The fructose:glucose ratio in honeydew honey is normally higher than that in nectar honeys. In addition, melecytosis is a sugar which is found exclusively in honeydew honey [42].

A less frequently used method is the titration technique in which methylene blue is used as an internal indicator. The difference in invert sugar concentrations before and after inversion, multiplied by a factor of 0.95, yields sucrose content [2].

Other methods recommended for determining sugar content in bee honey include gas chromatography (GC) and HPLC with pulsed amperometric detection. In the GC method, sugars are silylated and then the derivative fraction is quantified. Mannitol is used as an internal standard. The latter method (HPLC) is based on the principle that at high pH levels sugars behave like very weak acids—they are fully or partially ionized, and therefore they can be separated using the ion exchange mechanism [2].

A novel method for determining the content of fructose, glucose, and sucrose in bee honey samples has been proposed by scientists from Brazil [43]. Sugar content in their study was determined by capillary electrophoresis. The advantages of this method include high resolution, short sample preparation time, a small sample size, and short duration of the analysis itself. Within two minutes, the three compounds tested were separated completely, ensuring repeatability and linearity. The detection limits were 0.022–0.029 g/L.

High performance thin layer chromatography with image analysis was used by Puscas et al. [44] to detect “adulterants” in bee honey. This technique allowed the authors to analyze basic sugars present in honey: fructose, glucose, and sucrose. Moreover, the authors calculated the fructose:glucose ratio. The study demonstrated that acacia honey was the most faked honey. The reason was probably the producers’ desire to improve the organoleptic characteristics of this variety. The advantage of this method is its simplicity and low cost of analysis. Therefore, it can be an alternative to more expensive and more time-consuming techniques used to detect adulteration [44].

A different method of analyzing the content of glucose, fructose, sucrose, and additionally, maltose, has been proposed by researchers from Turkey [45]. It is based on Raman spectroscopy, followed by advanced statistical techniques of PLS, PCA, and ANN. The authors obtained a high correlation coefficient between the values determined and those predicted by the models.

Another technique that enables the determination of monosaccharide and disaccharide content is surface-assisted laser desorption with mass ionization spectrometry using HgTe nanostructures as a matrix (SALDI-MS). Sucralose is used as the standard. The authors emphasize that this method does not require time-consuming sample preparation and analysis time is only 30 min. In addition, it is characterized by high repeatability [46].

Ma et al. [47] developed a method for assessing honey maturity on the example of 85 acacia honeys for which 18 physicochemical parameters were analyzed in combination with chemometric analysis. During the process of maturation, honey changes its chemical composition. Mature honey has a moisture content below 18% and sugar concentration above the saturation point—it is enclosed in honeycomb cells. The main differentiating variables revealed in the analysis of variance were total sugar content (fructose, glucose, and sucrose), total protein content and total phenolic content. PCA, CA, and orthogonal partial least squares-discriminant analysis (OPLS-DA) were used in the classification.

### 3.3. Water

The harvesting of unripe honey will result in it having too high a water content. This will cause faster fermentation as a result of microbial development, including *Zygosaccharomyces*. Water, by collecting in higher layers of honey causes thinning, followed by foaming, an acidic smell, and a characteristic taste. On the other hand, yeast from the genus *Torulopsis* cause fermentation which is manifested, for example, by honey leaking from its packaging [42]. The susceptibility of honey to the development of microorganisms increases in samples with a water content above 17% [40].

Moisture content in honey is measured by refractometry. This parameter is determined in honey melted at 50 °C and then cooled to room temperature. The principle of this method is founded on measurement of the refractive index, based on which water content in honey is determined. The higher the solids content, the higher the refractive index. The result is expressed in g/100 g [2]. Requirements in European Union countries are based on the regulation that the water content in honey should be no more than 20 g/100 g with the exception of baker’s and heather (*Calluna* (L.) Hull.) honey (the norm: no more than 23 g/100 g) and baker’s honey from heather (the norm: no more than 25 g/100 g) [1].

New methods of assessing moisture content in honey are based on evaluating the way of crystallization by studying crystal size and shape. Tappi et al. [48] utilized Differential scanning calorimetry (DSC) and time domain magnetic resonance (TD-NMR) methods. The TD-NMR method allowed the authors to distinguish two pools of protons, whose relative intensity was approximately 55% and 45%. It was also observed that static crystallization is divided into two stages, with the second partially reversing the effects of the first. The study confirmed that in a dynamic process with continuous stirring, crystallization time of honey is reduced 5–6 times [48].

The fructose: glucose ratio affects the speed and manner of honey crystallization. In exemplary honeys from India studied by Naik et al. [49], the ratio was 0.931, 1.17, 1.18, 1.23, and 1.54. The most stable crystals were formed with the fructose: glucose ratio of 1.18, which was confirmed by simulations using ANN. Other ingredients that influence the manner of crystallization are maltose, sucrose, and water. This study proves that the recommended ratio (1.18) allows for a delay or even avoidance of the crystallization process. It also enables us to understand the interactions of sugars present in honey using molecular dynamics [49].

### 3.4. Enzymes: Diastase and Invertase

Enzymes present in honey include invertase, amylases (α- and β-amylase), maltase, phosphatases, catalase, glucose oxidase [40], β-fructofuranosidase, and ascorbinoxidases [42]. What is particularly important in regard to enzymes, their content in bee honey is influenced by, among others, storage conditions (including high temperatures), and decrystallization process [40].

A unit of diastase activity is the amount of enzyme contained in bee honey that converts 0.01 g of starch (insoluble starch conjugated with blue dye is used as a substrate) within 1 h at 40 °C. Determination is performed by spectrophotometric reading at 620 nm. The result is expressed in units of Schade/g honey [2]. Good quality honey should have no less than 8 Schade units/gram (except baker’s honey). Honeys with a low enzyme content (citrus honeys with an HMF content of no more than 15 mg/kg) can have a diastase number of no less than 3 [1].

One unit of invertase activity is the number of micromoles of a substrate that are destroyed per minute. The substrate used is p-nitrophenyl-α-D-glucopyranoside, which breaks down into glucose and p-nitrophenol under the influence of α-glucosidase (also called invertase). At pH 9.5, the reaction is trapped and the nitrophenol is converted to the nitrophenol anion—the amount of a substrate is determined based on its amount. Spectrophotometric reading is taken at 400 nm [2].

An alternative method for determining diastatic activity of bee honey has been developed by Sakač & Sak-Bosnar [50]. The proposed method uses a platinum redox sensor and is based on the potentiometric measurement of free triiodide, which is released from the triiodide–starch complex during degradation. The authors compared the new, fast measurement technique with the Schade method and the method based on incubation with Phadebas tablets. For most samples, there were no significant differences in activity determination between the three methods studied. Moreover, the analysis time of one sample in the proposed technique is only 5 min, and therefore much shorter than, e.g., that of the aforementioned method of incubation (where incubation time is around 60 min).

In 2019, a paper on the determination of diastase number by Visable-Near Infrared (VIS/NIR) spectroscopy was published. The Gaussian Filter Smoothing-standard Normal Variate (GF-SNV) and least squares-support vector machine (LS-SVM) were used. The root-mean-square error (RMSE) was 0.2 [51].

In addition, the enzymes β-fructofuranosidase and β- and γ-amylase are used in the production of syrups from starch and therefore their high activity indicates the presence of sugars originating from starch syrups in honey [52].

### 3.5. pH and Free Acidity

Organic acids, present in bee honey, are the key factor responsible for its taste. Among the most common organic acids are citric and gluconic acid, as well as succinic, malic, butyric, lactic, formic, acetic, and pyroglutamic acids. These acids affect the overall acidity of honey (called titration) [40].

The negative logarithm of hydrogen ion concentration is pH, while free acidity is the sum of all free acids present in honey. The principle of the method is based on the dissolution of bee honey in water, pH measurement, followed by titration with sodium hydroxide solution (0.1 M) to obtain a pH of 8.3 [2]. Free acidity should be no be more than 50 milliequivalents/kg of honey with the exception of baker’s honey (the norm: no more than 80 milliequivalents/kg) [1].

Acidity level and water content in honey are parameters that affect the development of yeast and mold in this bee product. For example, an analysis of honeys from Brazil, published in 2014, showed that 45.7% of honeys were of inferior quality [53].

With regard to pH and free acidity determination, no recent data on faster methods for analyzing these parameters is available. As far as free acidity is concerned, newer techniques would make analysis easier as titration has to be performed quickly—within 2 min—which, in laboratory practice, is not always successful and requires the analysis to be repeated several times, which leads to the destruction of valuable samples. An innovative approach in the case of these parameters consists in data presentation. In 2020, Ratiu et al. published a study in which correlations between the analyzed parameters, including pH, were shown using heat maps [54].

### 3.6. Ash and Elements

Mineral substances, amino acids, and organic acids (e.g., citric acid), present in bee honeys, form ionic forms in honey aqueous solutions, which consequently affects the conduction of electrical current and the measurable parameter referred to as electrical conductivity [40]. Minerals, after burning honey, remain as ash [42].

Determining ash content is helpful in assessing the type of honey. Ash is the residue after burning a sample at 350–400 °C, for at least 1 h, repeated until a constant weight is obtained. Its amount is expressed in g/100 g [2].

Electrical conductivity is determined on a 20% honey solution, calculated on dry matter. Determination of electrical conductivity is based on the measurement of electrical resistance, which is the inverse of conductivity. The result is expressed in mS/cm [2]. This parameter should be no more than 0.8 mS/cm with the exception of, among others, honeydew honey [1].

Newer methods allow for detailed examination of the composition of honey, including the content of macroelements, microelements, and toxic elements.

The capillary electrophoresis method with UV detection used by Shi et al. [55] determined 11 metal cations present in bee honeys (Ba^2+^, Ca^2+^, Cd^2+^, Cr^3+^, Cu^2+^, Fe^2+^, K^+^, Li^+^, Mg^2+^, Na^+^, and Zn^2+^), which affected the quality of bee honey. Soil and bee food are mentioned as sources of cations. The varied composition can be the basis for distinguishing honey varieties as well as for safety assessment and detection of adulteration. The advantages of this technique include a short analysis time (around 8 min) and a detection limit of 0.01–0.21 mg/L [55].

The determination of 14 elements (Ca, Cd, Co, Cr, Cu, Fe, K, Mg, Mn, Na, Ni, P, Pb, and Zn) in honeys obtained from Poland and various European regions (*n* = 66) was conducted by Grembecka and Szefer [56]. Statistical analyses were conducted resulting in dendrograms discriminating dark (heather, buckwheat, and honeydew) from light honey (rape, linden, acacia, and multifloral). This confirmed a higher content of elements in dark honeys.

The content of lead, cadmium, and chromium was determined by scientists from Brazil in 2014 by Electrothermal Atomic Absorption Spectrometry (ET-AAS) technology [57]. Determining the concentration of these elements in honey can be useful as a bioindicator of environmental contamination. The advantages of this method, in relation to honey analysis, include the following: no need to pre-treat samples, precision and accuracy of determination. In addition, honeys can act as a bioindicator of environmental pollution in regard to radioactive elements [58,59].

The content of microelements and trace elements in bee honey was the subject of research by Ribeiro et al. [60]. The authors used the Total Reflection X-ray Fluorescence Spectroscopy (TXRF) technique. Honeys were obtained in 4 seasons: spring, summer, autumn, and winter. Seasonal changes in potassium (e.g., winter: 98.7 mg/100 g ± 12, autumn: 194.5 mg/100 g ± 20) and calcium (e.g., winter: 375.2 mg/100 g ± 38, autumn: 46.4 mg/100 g ± 6) levels were demonstrated. However, no significant changes were found in the content of Cr, Ti, Se, and Ni.

### 3.7. Proline

Proline is the major amino acid in bee honey, constituting 49% in nectar honey and 59% in honeydew honey of the total amino acid content. Other important amino acids present in nectar and honeydew honey include phenylalanine (32% and 6%) and glutamic acid (3% and 11%), respectively [40].

Determination of proline content in bee honey is performed, among others, to confirm the authenticity of honey. The content of this amino acid is determined on the basis of its colored complex formed with ninhydrin; absorbance is read at 510 nm [2]. The proline content of honey should be no less than 25 mg/100 g [42].

The assessment of amino acid or protein content in bee honey and use of these ingredients by bees is the subject of modern research.

Proteins in honey constitute from 0.1 to 0.5%. The main ones are invertase (α-glucosidate) and MRJP. To date, methods such as dialysis against distilled water and lyophilisation of the resulting dialysate have been used to isolate proteins as the so-called gold standard. Bocian et al. [61] developed a faster and cheaper extraction method using saturated phenol. It was tested on 3 types of honey: black locust, buckwheat, and rapeseed. The main advantages of the method are an almost threefold reduction in protein isolation time from 32 to 12 h and no need for expensive equipment (including a freeze dryer). Electrophoresis demonstrated that the proteomes of individual varieties differ from each other, which allows for distinguishing varieties of bee honey.

### 3.8. Polyphenols and Other Antioxidant Composition

Phenolic substances, which are phenol derivatives, are synthesized by plants. They are divided into simple phenols and polyphenols. Polyphenols are characterized by the presence of more than one aromatic ring in the structure of the molecule [62].

The principle of determining total TPC is as follows: phenolic compounds that are present in honey are oxidized under the influence of the Folin–Ciocalteu reagent. The components of this solution, phosphomolybdic acid, and phosphotungstic acid, are reduced to molybdenum and tungsten oxides, which yield a blue color. The color intensity is directly proportional to the content of phenolic compounds present in the sample [63]. Differences in the content of phenolic compounds in individual honey varieties were used by Zhang et al. [64]. The authors developed a method for the determination of chlorogenic, gallic, ferulic, caffeic, p-hydroxybenzoic, p-coumaric, protocatechuic, syringic acids, and rutin.

In 2014, a study investigating differences in the content of antioxidant compounds in honey samples was also published. The content of phenolic acids (benzoic, chlorogenic, gallic, coffee, and trans-cinnamic acids) as well as flavonoids (catechin, kemferol, myricetin, and naringenin) was determined [65]. The authors indicated that differences in compound concentration between individual varieties were caused, among others, by the botanical origin of honey. The highest levels of benzoic and caffeic acid were found in bee honeys from Bangladesh.

Both older and more recent publications are based on the color assessment method in which the absorbance of a 50% honey solution is measured at a wavelength of 635 nm [66]. Additionally, the color intensity of honey is assessed by measuring the color intensity ABS450—the absorbance of a 50% honey solution at 450 and 720 nm [67]. Many authors correlate the obtained data with other parameters, looking for dependencies and explaining the biological activity of honey. Publications from recent years that evaluate the antioxidant properties of honey are based on previously developed methodologies and focus on, inter alia, determining the content of various compounds including flavonoids, ascorbic acid, carotenoids, β-carotene, lycopene, reducing sugar, as well as assessing radical scavenging activity with 2,2-diphenyl-1-picrylhydrazyl (DPPH) [68], ferric-reducing/antioxidant power (FRAP) assay [69], and total antioxidant capacity (TAC) [70]. These methods are based on spectrophotometric absorbance measurements (Table 4). They are fairly fast and the obtained results are repeatable and therefore they constitute the basis of modern research on antioxidant properties. These properties are crucially important from the point of view of the prophylactic abilities of honey and their use in supporting the treatment of many diseases, which has been confirmed by numerous publications within the last 10 years [71,72,73].

### 3.9. Hydroxymethylfurfural

5-hydroxymethylfurfural is a component resulting from the Millard reaction. It is formed under the conditions of increased temperature during the reaction of dehydration of sugar. This reaction takes place in an acidic environment. There are also reports in the available literature regarding HMF formation during caramelization and degradation of hexoses [40].

The principle of the method is based on the measurement of 5-(hydroxymethyl-)furan-2-carbaldehyde content in honey by HPLC with reversed phase and UV detection. HMF content in the standard is determined by spectrophotometry at 285 nm. The result is expressed in mg/kg [2]. Honey available for sale may contain no more than 40 mg/kg, except for baker’s and tropical honeys (the norm: no more than 80 mg/kg) [1]. Other methods (White and Winkler) are less frequently used [2].

An alternative method for determining HMF content was proposed in 2013 by Hošt’alková et al. [74]. The authors used high-performance thin-layer chromatography (HPTLC) and the Reflectoquant spectrophotometric test. In both techniques, short analysis time (2.5 min) was obtained and deviation between the methods was 15%. The HPTLC method was characterized by higher precision. The advantage of both techniques is the absence of harmful reagents, which is in line with the principles of green chemistry.

NIR spectroscopy is gaining importance in food sample analysis. Stőbener et al. [75] used Fourier Transform Attenuated Total Reflection Infrared Spectroscopy (ATR FTIR) to determine the content of 5-hydroxymethylfurfural in bee honey. This compound is a cyclic aldehyde formed in the Millard reaction by dehydrating hexoses, catalyzed by acid, at a high temperature. The authors added a HMF solution to honey at a concentration of 9 to 100 ppm, every 5 ppm. HMF alone produces vibrating bands in the spectrum from 3500 to 2700 and from 1850 to 600 cm^−1^. After adding the HMF standard to honey, no vibrational characteristic of pure analyte was observed. The authors, using modelling and several calibration models, developed a method for determining HMF in honey. They used chemometric techniques such as non-linear iterative partial least squares (NIPALS). The authors observed that for the proper development of a method used to determine trace compounds, it is necessary to use a large number of calibration samples in a wide range of concentrations. The best model was a model based on spectrum averaging—it allowed to reduce noise and improve forecasting parameters. Using this technique, it is possible to detect the HMF concentration of 13 ppm in bee honey.

### 3.10. Insoluble Matter and Contaminants

Impurities in bee honey can enter the final product during preparation, centrifugation, or packaging. The source of insoluble substances may be an improperly performed process of straining honey during extraction from honeycombs. During this process, the bee product flowing from the honey extractor passes through a series of sieves [42].

Ingredients insoluble in water (at 80 °C), present in honey, constitute the residue left after the filtration of honey solution. The residue is dried at 135 °C, to a constant weight. The result is expressed in weight percent [2]. Honey should contain no more than 0.1 g/100 g of insoluble ingredients, except for pressed honey (the norm: no more than 0.5 g/100 g) [1].

Due to the prevalence of diseases causing massive extinction of bees, techniques for detecting undesirable residues of various compounds in natural bee honey are gaining popularity. Among xenobiotics, sulfonamide residues should be mentioned [76,77]. They are used to control 3 major threats to honeybees: European foulbrood caused by *Melissococcus plutonius*, American foulbrood, which is caused by *Paenibacillus larvae* ssp. Larvae and Varroa, which is based on *Varroa destructor*. However, the presence of these substances in bee products may present a risk to consumers [78]. Sajid et al. [77] performed determination of sulfonamide residues by extraction using a short C-18 column, utilizing high-performance liquid chromatography with fluorescence detection. The advantage of this method is its low cost (including reagent utilization), speed of execution and short extraction time [77]. Szczęsna et al. [79] demonstrated the presence of sulfonamide residues in honey in the range of 5–2891 µg/kg, revealing that 17% of tested honeys exceeded the permissible concentration limits.

High resolution mass spectrometry has also been used to detect xenobiotics, combined with data mining analyses and statistical tools useful in metabolomic techniques. A study by Cotton et al. [80] revealed that the concentrations of 35 compounds in tested honeys were above the permitted limits. This technique can also be used to classify honeys (acacia, orange, lavender, and multifloral).

Azzouz & Ballesteros [81] developed a method for determining 22 pharmacologically active substances present in honey (including non-steroidal anti-inflammatory drugs, antibacterial agents, β-blockers, lipid regulators, antiseptics, and anti-epileptics) by GC-MS, using the solid phase extraction (SPE) module. Some of these substances may be present in honey as a result of fighting honeybee brood diseases European, such as European brown rot. Therefore, the monitoring process of these substances seems necessary. The advantages of this method include accuracy, sensitivity, and low matrix interference, as evidenced by a recovery of 86–102%.

Pesticides are one of the factors responsible for weakening bees and increasing mortality of their populations [82]. The technique for determining 22 insecticides belonging to 3 chemical groups (neonicotinoids, pyrethroids, and pyrazoles) was developed by Paradis, Bérail, Bonmatin and Belzunces [83]. The study material were samples of rapeseed, acacia, chestnut, and multifloral honey. Most compounds (acrinathrin, bifenthrin, cypermethrin, deltamethrin, esfenvalerate, fipronil, fipronil desulfinyl, fipronil sulfide, fipronil sulfone, λ-cyhalothrin, permethrin, pyraclofos, resmetrin, tebufenpyrad, τ-fluvalinate, and tolfenpyrad) were determined by GC-MS/MS, while the remaining 6 were determined by liquid chromatography coupled with mass spectrometer (LC-MS/MS, liquid chromatography–mass spectrometry) (acetamiprid, clothianidin, ethiprole, imidacloprid, thiacloprid, and thiamethoxam). The advantage of this technique was the detection of pesticides which were present in honey in very low concentrations (below 1 ng/g).

The liquid chromatography technique with mass spectrometer and ion trap is also used to detect pyrrolizidine alkaloids (PAs). These alkaloids, which are secondary plant metabolites, have hepatotoxic effects. They are found in plants from the families Asteraceae, Fabaceae and Boraginaceae. Among the 11 alkaloids analyzed, lycopsamine (182–4078 µg/kg) was the most common [84].

A description of another technique, also used to detect pyrrolizidine alkaloids, was published in 2014. It was based on an enzyme-linked immunosorbent assay (ELISA). The authors developed a method whose detection capacity for Jacobin, heliotrine, lycopsamin, and sennesionin was 25 µg/kg. The advantages of this method include low cost, short analysis time (up to 2.5 h), and the possibility of testing 21 samples simultaneously [85].

Methods for detecting contaminants in honey are still evolving as confirmed by data collected in Table 5 which contains selected impurities detected in the last 10 years.

### 3.11. Adulterations

Due to the health-promoting properties of bee honey, the consumer demand for it has increased in recent years, which has resulted in the necessity to import honey from outside the European Union (EU). One way of falsifying bee honey is by adding cheaper HFCS. Bee honey is one of the most frequently falsified foods [107,108]. Çinar et al. [109] searched for a method of detecting adulteration of bee honey with HFCS by a method of calculating the ^13^C/^12^C isotope ratio. This proportion is influenced by the type of photosynthesis performed by plants: C3 (plants convert CO_2_ to a 3-carbon compound), C4 (conversion to a 4-carbon compound) and CAM (plants use both cycles) [110]. Representatives of the C4 group are, among others, corn plants, and sugar cane [108]. It has been shown that the higher the HFCS concentration, the higher the C4 sugars. Calculation of the amount of sugars in C4 plants, based on the determination of the ratio of ^13^C/^12^C isotopes in the C atom derived from CO_2_ by mass spectroscopy, is the basis for detecting adulteration of HFCS honey [109].

Research into honey adulteration based on the ^13^C/^12^C isotope ratio is continuing as evidenced by a study published in 2019 by authors researching honeys from Taiwan [111].

An innovative method was developed to identify this type of adulteration based on head space-ion mobility spectrometry (HS-IMS). For testing purposes, honey samples were adulterated with syrup in the range of 10 to 50%. The following statistical methods were used: HCA and PCA. The advantage of this method is low detection limit, ability to monitor the amount of the adulterant in real time as well as absence of toxic solvents and low gas consumption, which is consistent with the principles of green chemistry. The authors optimized 5 parameters: incubation time (15 min), incubation temperature (50 °C), sample size (0.15 g honey), volume of injection (0.91 mL), and heating time (22 min) [112].

The technique of three-dimensional fluorescence, associated with multidimensional calibrations, was used to determine the concentration of rice syrup in bee honeys and thus to assess adulteration with these syrups [113]. The analyses were conducted, among others, on rape, buckwheat, and sunflower honey samples. Statistical methods such as PCA, PLS, and ANN were used. The aim was to develop an accurate, fast, and non-invasive method of detecting adulteration.

The HPLC technique is used to detect adulteration with starch syrup [114]. A characteristic peak with a retention time of 15.25 min, derived from starch syrup, is observed. The authors of the method emphasize its low cost and simplicity of implementation.

The methods for detecting adulteration of honey are constantly being improved. In 2019, a paper on the use of VIS-NIRS in combination with chemometric methods was published. Counterfeiting was performed with the following substances: rice syrup, brown cane sugar, fructose syrup, and inverted sugar in concentrations from 5 to 50%. Full distinction between adulterated and unadulterated multifloral honey was obtained by the LDA (Linear Discriminant Analysis) method [115].

## 4. Conclusions

Methods enabling the examination of several parameters which would prove the quality of bee honey within a short period of time are needed. Modern techniques which have been developed and validated over the past few years provide high precision and accuracy but need further improvement so that the natural bee honey available on the market can be tested as quickly and reliably as possible. Future research into methods of honey quality assessment should focus on developing techniques of rapid evaluation of the botanical origin of honey since the principal method currently in use, the melissopalynological method, is very time consuming, requires considerable experience, botanical knowledge as well as knowledge of the honey production process. It produces inconclusive results which can be difficult to interpret. Moreover, it is necessary to devise quick, reliable, and inexpensive methods for detecting honey adulteration. As an overview of current methods of honey quality assessment, this paper can help shape the direction of future research in this field.

## Figures and Tables

**Figure 1 foods-09-01028-f001:**
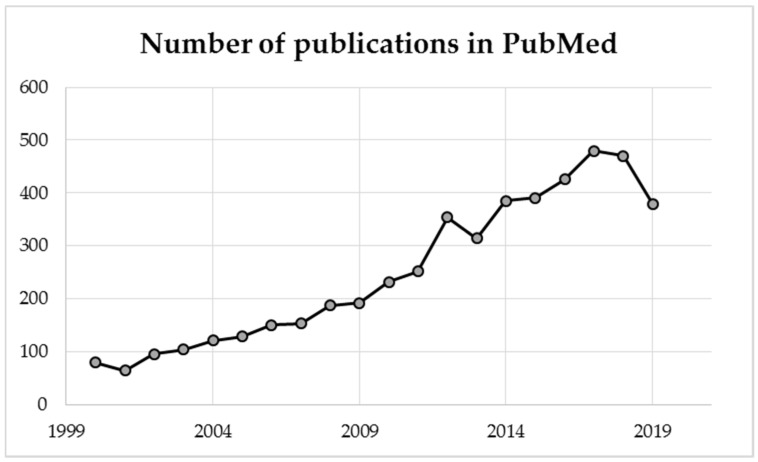
Number of publications in the PubMed database with the word “honey” in the title.

**Table 1 foods-09-01028-t001:** Composition of honeydew and blossom honey [20].

Component	Honeydew Honey	Blossom Honey
	(Min−Max)	(Min−Max)
	[g/100 g]	[g/100 g]
Fructose	31.8	38.2
Glucose	26.1	31.3
Water	16.3	17.2
Sucrose	0.5	0.7
Other disaccharides	4.0	5.0
Melezitose	4.0	<0.1
Erlose	1.0	0.8
Other oligosaccharides	13.1	3.6
Acids	1.1	0.5
Minerals	0.9	0.2
Amino acids, proteins	0.6	0.3

**Table 2 foods-09-01028-t002:** Selected methods for determining type and variety of honey.

Method	Country	Varieties of Honey or Origin	Literature
Electronic with UV—VIS—NIR	Portugal	*Arbutus unedo* L.—strawberry-tree (*n* = 4), *Citrus* spp.—orange blossom (*n* = 3), *Helianthus annuus* L.—sunflower (*n* = 3), *Lavandula stoechas* L. — French lavender (*n* = 3)	[23]
Electronic tongue	Croatia	*Castanea sativa* Mill.—chestnut (*n* = 16), *Robinia pseudoacacia* L.—black locust (*n* = 49)	[24]
Electronic potentiometric tongue	Portugal	Representative samples from regions: Algarve regions, Alentejo, Beira Interior, Beira Litoral, Entre Douro e Minho, Estremadura e Ribatejo, Trás-os-Montes e Alto Douro, and Pico and São Miguel islands (*n* = 65)	[25]
Electronic nose	Poland	acacia flower (*n* = 3), buckwheat (*n* = 3), honeydew (*n* = 3), linden flower (*n* = 3), and rape (*n* = 3)	[26]
NMR	Bulgaria	citrus (*n* = 1), fir honeydew (*n* = 1), honeydew (*n* = 1), oak honeydew (*n* = 15), polyfloral (*n* = 4), rapeseed (*n* = 1), and spruce honeydew (*n* = 1)	[27]
Twodimensional zymography	Italy	*Castanea sativa*—chestnut (*n* = 4), *Citrus*—orange (*n* = 4), *Eucalyptus* sp.—eucalyptus (*n* = 4), and *Hedysarium coronarium*—sulla (*n* = 4)	[28]
PCR	Australia, Brazil, China, South Africa Vietnam	*Apis carena:* China (*n* = 3), Vietnam (*n* = 3), *Apis mellifera*: Australia (*n* = 3), Brazil (*n* = 3), China (*n* = 31), and South Africa (*n* = 2)	[29]
CIE L * C * abh ab scale	Croatia, France, Germany, Hungary, Italy, Poland, Spain, Ukraine	*Arbatus unedo* L.—strawberry tree (*n* = 42), *Asphodelus microcarpus* Salzm. Et Viv—asphodel (*n* = 36), *Brassica napus* L.—rapeseed (*n* = 14), *Castanea sativa* Mill.—sweet chestnut (*n* = 14), *Citrus* spp.—citrus (*n* = 9), *Erica* spp.—heather (*n* = 11), *Eucalyptus* spp.—eucalyptus (*n* = 22), *Fagopyrum esculentum* L.—buckwheat (*n* = 12), honeydew (*n* = 22), *Galactites tomentosa* Moench—thistle (*n* = 26), *Hedysarum coronarium* L.—sulla flower (*n* = 16), *Mentha* spp.—mint (*n* = 12), *Paliurus spina-christi* Mill.—garland thorn (Christ’s thorn) (*n* = 14), *Robinia pseudoacacia* L.—black locust (*n* = 14), *Tilia* spp.—lime (*n* = 12), *Salvia officinalis* L.—sage (*n* = 14), *Satureja* spp.—savory (*n* = 15),	[30]
Fluorescence spectroscopy	Serbia	acacia (*n* = 37), fake honey (*n* = 14), linden (*n* = 10), meadow mix (*n* = 23), and sunflower (*n* = 11).	[31]
DNA identification and plastids	Italy	Multifloral (*n* = 4)	[32]
DNA metabarcoding	USA	Pollen from hives (*n* = 4)	[33]
Statistical analysis: PLS-DA	China	Acacia from six geographical regions of China: Gansu (*n* = 10), Henan (*n* = 12), Liaoning (*n* = 10), Shaanxi (*n* = 14), Shandong (*n* = 15), and Shanxi (*n* = 10)	[34]
Continuous flow mass spectrometry		Hive frames with 64 seats	[35]

CIE L * C * abh ab scale—Commission Internationale de l’Eclairage L * C * abh ab scale, UV-VIS-NIR—Ultraviolet-Visable-Near Infrared.

**Table 3 foods-09-01028-t003:** Markers for selected varieties of honey.

Honey Varieties	Botanical Origin	Percent of Specific Pollen	Markers	Method	Literature
Buckwheat	*Fagopyrum esculentum* L.	32–53	3-hydroxybenzoic acid; ferulic acid	LC-DAD, GC-MS	[30]
Citrus	*Citrus* spp.	14–39	caffeine; methyl anthranilate	LC-DAD, GC-MS	[30]
Heather	*Erica* spp.	40–62	4-methoxybenzaldehyde; 4-methoxybenzoic acid; methyl 4-hydroxy-3-methoxybenzoate	GC-MS	[30]
Lime	*Tilia* spp.	11–47	1-(4-methylphenyl)ethanone; 4-terpinenol	GC-MS	[30]
Mint	*Mentha* spp.	18–39	methylnsyringate; vomifoliol	LC-DAD, GC-MS	[30]
Rape	*Brassica napus* L.	-	kaempferol; morin	HPLC-MS, with tandem ion detect	[38]
Toran, Saha	-	-	2-amino-4-hydroxypteridine-6-carboxylic acid, methyl 3-hydroxyhexanoate	GC-MS	[39]

GC-MS—gas chromatography mass spectrometry, HPLC-MS—high performance liquid chromatography mass spectrometry, LC-DAD—liquid chromatography with diode array detection.

**Table 4 foods-09-01028-t004:** Selected methods of assessing the antioxidant properties of honey.

Method	Wavelength	Unit	Literature
Ascorbic acid	515 nm	mg ascorbic acid/kg	[68]
Carotenoids content	453, 505, and 663 nm	mg of carotenoid/kg	[68]
Color intensity ABS_450_	450 and 720 nm	mAU	[67]
Color in Pfund scale	635 nm	mm	[66]
DPPH (scavenging activity)	517 nm	% radical scavenging activity, ICE50	[67]
Flavonoid contents	510 nm	mg (+)-catechin equivalents /kg	[68]
FRAP assay	593 nm	μM Fe(II)	[67]
ORAC	Emission: 535 nm Excitation: 485 nm	trolox equivalent/g	[67]
Phenol content	750 nm	mg gallic acid/kg	[67]
Reducing power	700 nm	EC50	[68]
TAC	695 nm	Ascorbic acid equivalents/g or gallic acid equivalents/g	[70]

DPPH—2,2-diphenyl-1-picrylhydrazyl, FRAP—reducing/antioxidant power assay, ORAC—oxygen radical absorbance capacity, TAC—total antioxidant capacity.

**Table 5 foods-09-01028-t005:** Selected contaminants detected in bee honey over the last 10 years.

Component	Year of Publication	Literature
aminoglycosides	2015	[86]
aflatoxin	2013	[87]
*Bacillus subtilis* and *Bacillus cereus*	2014	[88]
benzimidazole derivatives	2015	[89]
bromine and iodine	2015	[90]
cadmium	2019	[91]
*Candida lundiana* sp. nov. i *Candidia suthepensis* sp. nov.	2012	[92]
cannabinoids	2019	[93]
chloramphenicol	2013	[94]
chlorophenol	2015	[95]
cobalt	2012	[96]
fluoroquinolones	2013	[97]
lead	2014	[98]
lincomycin	2014	[99]
mayanotoxin from *Rhododendron ponticum*	2014	[100]
mercury	2012	[101]
metronidazole	2011	[102]
nickel	2015	[103]
penicillin	2013	[51]
polycyclic aromatic hydrocarbons	2012	[104]
sulfonamides	2012	[105]
tetracycline	2015	[106]
^137^Cs	2013	[59]

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
