# Peer review of "Modern Methods for Assessing the Quality of Bee Honey and Botanical Origin Identification"

_foods, 2020, doi:10.3390/foods9081028_

Round 1

Reviewer 1 Report

The manuscript entitled “Modern methods for assessing the quality of bee honey and botanical origin identification” needs to be revised. Some recommendations are as the following:

- A table summary composed of the major important studies could be very useful as a narrative review harder to follow (especially for section 3.1).

- Line 132: Here it might be useful to explain PDO. For that, the authors may refer to the following publication: doi: 10.1016/j.foodchem.2018.10.09.

- The conclusion can be extended. Please make recommendations for future research.

- In addition to the above, the manuscript requires minor language revision. Please check the entire manuscript carefully.

Author Response

Dear Reviewer Please see the attachment.

COVER LETTER

Reply to the Reviewer 1

We thank the reviewer for taking the time to read our work. We tried to take into account all suggestions. Below are the reviewer's recommendations and our responses.

- A table summary composed of the major important studies could be very useful as a narrative review harder to follow (especially for section 3.1).”

As suggested, we added Table 2 - it begins in verse 116,

Table 2. Selected methods for determining the type and variety of honey.

Method

Literature

Electronic tongue with UVA – VIS – NIR

Electronic tongue

Electronic potentiometric tongue

Electronic nose

NMR

[23]

[24]

[25]

[26]

[27]

Twodimensional zymography

PCR

CIE L * C * abh ° ab scale

Fluorescence spectroscopy

[28]

[29]

[30]

[33]

DNA identification and plastids

[34]

DNA metabarcoding

[35]

Statistical analysis: PLS-DA

[36]

Continuous flow mass spectrometry

[37]

- Line 132: Here it might be useful to explain PDO. For that, the authors may refer to the following publication: doi: 10.1016/j.foodchem.2018.10.09.

We introduced the abbreviation PDO in the text. We believe that the suggested DOI publication number is shorter by the last digit, but to our knowledge we found the publication and added it to the text. We also added a source article on honey adulteration (verses 172-184)

Attempts have been made to distinguish between bee honey varieties using the CIE scale (Commission Internationale de l'Eclairage) L * C * abh ° ab [30]. The authors Tuberoso et al. assessed the colour of 17 monofloral honeys (n = 305). The advantage of the proposed method is small sample size (approximately 2 g) and absence of any destructive effect (samples can be reused in subsequent analyses). Principal Component Analysis (PCA) and hierarchical clustering analysis (HCA) have been used to distinguish between different honey varieties and classifications. In the case of reflection methods, the L * value for light honeys is below 50 while for dark honeys, it is above 50. The authors emphasise the fact that this technique may be particularly useful in the case of honeys with the protected designation of origin (PDO) mark. This concept covers honeys produced, processed and prepared in one area which have distinct characteristics from this area. Their names are legally protected and listed on the EU protected food name register [1]. The development of rapid tests which would assist in honey identification is particularly important since bee honey is one of the 25 products at highest risk of adulteration [31,32].

- The conclusion can be extended. Please make recommendations for future research.

As suggested, we expanded the conclusions - they are currently in the text from verse 592 to 602,

Methods enabling the examination of several parameters which would prove the quality of bee honey within a short period of time are needed. Modern techniques which have been developed and validated over the past few years provide high precision and accuracy but need further improvement so that the natural bee honey available on the market can be tested as quickly and reliably as possible. Future research into methods of honey quality assessment should focus on developing techniques of rapid evaluation of the botanical origin of honey since the principal method currently in use, the melisopalinological method, is very time consuming, requires considerable experience, botanical knowledge as well as knowledge of the honey production process. It produces inconclusive results which can be difficult to interpret. Moreover, it is necessary to devise quick, reliable and inexpensive methods for detecting honey adulteration. As an overview of current methods of honey quality assessment, this paper can help shape the direction of future research in this field.

- In addition to the above, the manuscript requires minor language revision. Please check the entire manuscript carefully.

We corrected some minor editing errors (we wrote a few Latin names in italics). Additionally, the text of the manuscript was checked by a native speaker to eliminate errors.

Thanks again for any suggestions. We hope that the changes we have introduced will increase the substantive value of the work.

Yours faithfully,

Authors

Yours faithfully,
Authors

Reviewer 2 Report

The type of paper should be changed from "Review" to " Perspective". The abstract should be better summarized. The aim should explain the aim of this work and the novelty character.  In the Conclusion limits, advantages , novelty and practical applications should be discussed. 

Lines 31-39 should be enlarged.

The paragraph 2. Honey composition is very poor! Example of previous work reporting honey composition and factors influencing honey quality should be inserted.

Lines 127-137 should be clarified. data in Table 2 should be better discussed and major details should be added in table 2.

Lines 187-188 should be enlarged.

Lines 237-243 should be better explained.

The paragraph 3.5. pH and free acidity should be implemented.

A paragraph on generally bioactive compounds and antioxidant properties should be added.

Author Response

Dear Reviewer Please see the attachment.

COVER LETTER

Reply to the Reviewer 2

We thank the reviewer for taking the time to read our work. We tried to take into account all suggestions. Below are the reviewer's recommendations and our responses.

- The type of paper should be changed from "Review" to " Perspective".

As suggested, we changed the manuscript type - verse 1

- The abstract should be better summarized.

To summarize the abstract better, we have added verses 25-28:

These methods are constantly modified, so that the honey that is on sale is a product of high quality. Prospects for devising methods of honey quality assessment include the development of a fast and accurate alternative to the melisopalinological method as well as quick tests to detect adulteration.

- The aim should explain the aim of this work and the novelty character.

To better define the aim, we have rewritten verses 63-66:

The aim of the present study was to summarise the most important methods of assessing honey quality published in the last 10 years. The novelty of these publication lies in the collection of data regarding both the evaluation of honey composition as well as adulteration.

- In the Conclusion limits, advantages , novelty and practical applications should be discussed.

To summarize better, we have added sentences about further perspectives and the need for research (verses 592-602),

Methods enabling the examination of several parameters which would prove the quality of bee honey within a short period of time are needed. Modern techniques which have been developed and validated over the past few years provide high precision and accuracy but need further improvement so that the natural bee honey available on the market can be tested as quickly and reliably as possible. Future research into methods of honey quality assessment should focus on developing techniques of rapid evaluation of the botanical origin of honey since the principal method currently in use, the melisopalinological method, is very time consuming, requires considerable experience, botanical knowledge as well as knowledge of the honey production process. It produces inconclusive results which can be difficult to interpret. Moreover, it is necessary to devise quick, reliable and inexpensive methods for detecting honey adulteration. As an overview of current methods of honey quality assessment, this paper can help shape the direction of future research in this field.

- Lines 31-39 should be enlarged.

In order to expand the paragraph better, we have added information on the types of honey and quality standards (verses 32-47)

Natural bee honey is a sweet product made by honeybees Apis mellifera L. both from the nectar of plants and from the excreta of insects sucking the juice from living parts of plants or from secretions of live parts of plants. Those are then combined with specific secretions of bees, stored in honeycombs, evaporated and left in honeycombs to mature. The Council of the European Union distinguishes nectar and honeydew honey as well as filtered honey, pressed honey, extracted honey, drained honey, comb honey and chunk honey or cut comb honey [1].

Bee honey intended for human consumption should be of sufficiently high quality. Requirements in individual countries include parameters such as variety determination, water content (no more than 20%; no more than 23% in heather and baker’s honey; no more than 25% in baker’s honey from heather), HMF (in principle, no more than 40 mg/kg), proline (no less than 25 mg/100 g), diastase activity (in principle, no less than 8 on the Schade scale), electrical conductivity (e.g. in honeydew honey no less than 0.8 mS/cm), pH, insoluble impurities (no more than 0.1 g/100 g) and free acidity (in principle, no more than 50 milli-equivalents acid/1000 g). Other parameters include, for example, colour and Total Phenolic Content (TPC). To assess honey quality, standard methods, including spectrophotometric, refractometric, titration and melisopalinological methods, are used [2].

- The paragraph 2. Honey composition is very poor! Example of previous work reporting honey composition and factors influencing honey quality should be inserted.

The composition we presented was very general, so we expanded it as suggested (verses 78-97).

Twenty five different sugars are found in honey. In nectar honeys, the sugars include erlose, maltose, sucrose and turanose, while in honeydew honey – melezitose and raffinose. Dextrins are found in Italian Metcalfa honey. Among the acids in honey, there is mainly gluconic acid, and also in smaller amounts: acetic, citric, formic, lactic, maleic, malic, oxalic, pyroglutamic and succinic acids. These acids influence the pH of honey, which is commonly between 3.3 and 4.6. Among the main minerals in honey, potassium should be mentioned (it constitutes approximately 33% of all elements present in honey). Others include barium (from 0.01 to 0.08 mg/100 g), boron (from 0.05 to 0.3 mg/100 g), chlorine (from 0.4 to 56 mg/100 g), cobalt (from 0.1 to 0.35 mg/100 g), fluoride (from 0.4 to 1.34 mg/100 g), iodine (from 10 to 100 mg/100 g), lithium (from 0.225 to 1.56 mg/100 g), molybdenum (up to 0.004 mg/100 g), nickel (up to 0.051 mg/100 g), rubidium (from 0.04 to 3.5 mg/100 g), silicium (from 0.05 to 24 mg/100 g), strontium (from 0.04 to 0.35 mg/100 g), sulfur (from 0.7 to 26 mg/100 g), vanadium (up to 0.013 mg/100 g) and zirconium (from 0.05 to 0.08 mg/100 g). The elemental composition is influenced by the origin – honeydew honeys are characterised by a higher content of elements. The most important and abundant amino acid in honey is proline. Moreover, enzymes such as amylase (diastase), invertase, catalase and glucose oxidase are present in honey. Enzymatic content proves the freshness of honey, and proper heating and storage conditions. Volatile compounds, both from the nectar and bee secretions, determine the smell of honey. Over 600 of them have been identified to date. By way of illustration, dark honeys contain more flavonoids. When honey is stored at too high a temperature and in the process of prolonged heating (particularly honeys with a lower pH), the content of the unfavorable component, HMF, increases [20].

- Lines 127-137 should be clarified. data in Table 2 should be better discussed and major details should be added in table 2.

In order to better understand the methods we describe, we have developed the characteristics (verses 164-193). Additionally, in Table 3, we have added information on the botanical origin of the plants and the share of predominant pollen.

The MRJP2 gene has also been used to discriminate between two types of honey – honey produced by Apis mellifera and that produced by Apis cerana. The latter is far more expensive and therefore there have been cases of mislabelling honey in terms of origin. In order to identify them correctly, the authors designed two pairs of species-specific primers. The amplification products of A. mellifera and A. carena honeys were 560 and 212 bp, respectively. The obtained primers were characterised by high species specificity. The MRJP2 gene was detected using the PCR method and selected primers. Differences in the gene formed the basis for establishing the origin of honey. The PCR method enabled detection of the addition of A. mellifera honey which was as low as 1% [29].

Attempts have been made to distinguish between bee honey varieties using the CIE scale (Commission Internationale de l'Eclairage) L * C * abh ° ab [30]. The authors Tuberoso et al. assessed the colour of 17 monofloral honeys (n = 305). The advantage of the proposed method is small sample size (approximately 2 g) and absence of any destructive effect (samples can be reused in subsequent analyses). Principal Component Analysis (PCA) and hierarchical clustering analysis (HCA) have been used to distinguish between different honey varieties and classifications. In the case of reflection methods, the L * value for light honeys is below 50 while for dark honeys, it is above 50. The authors emphasise the fact that this technique may be particularly useful in the case of honeys with the protected designation of origin (PDO) mark. This concept covers honeys produced, processed and prepared in one area which have distinct characteristics from this area. Their names are legally protected and listed on the EU protected food name register [1]. The development of rapid tests which would assist in honey identification is particularly important since bee honey is one of the 25 products at highest risk of adulteration [31,32].

Another method of honey classification is fluorescence spectroscopy and statistical analyses based on parallel factor analysis (PARAFAC) and partial least squares method combined with discriminant analysis (DA PLS) [33]. It was used in a study by Lenhardt et al, which investigated acacia, sunflower, lime, meadows and artificial honey (n = 95). Fluorescence in the range of excitation (260– 290 nm) and emission (330–360 nm) comes from aromatic amino acids and is particularly important in differentiating the botanical origin of honey samples. The authors found that phenolic compounds and Maillard reaction products had the greatest impact on the distinction of individual varieties – emissions of these compounds differed most between honey varieties. Classification efficiency was approximately 90%. Additionally, artificial honey was detected 100% correctly.

- Lines 187-188 should be enlarged.

We expanded this paragraph as suggested (verses 250-255)

Other methods recommended for determining sugar content in bee honey include Gas Chromatography (GC) and HPLC with Pulsed Amperometric Detection. In the GC method, sugars are silylated and then the derivative fraction is quantified. Mannitol is used as an internal standard. The latter method (HPLC) is based on the principle that at high pH levels sugars behave like very weak acids – they are fully or partially ionised and therefore they can be separated using the ion exchange mechanism [2].

- Lines 237-243 should be better explained.

We have extended the description to better understand the methods under discussion (verses 203-316)

New methods of assessing moisture content in honey are based on evaluating the way of  crystallisation by studying crystal size and shape. Tappi et al. utilised Differential Scanning Calorimetry (DSC) and time domain magnetic resonance (TD-NMR) methods. The TD-NMR method allowed the authors to distinguish two pools of protons, whose relative intensity was approximately 55% and 45%. It was also observed that static crystallisation is divided into two stages, with the second partially reversing the effects of the first. The study confirmed that in a dynamic process with continuous stirring, crystallisation time of honey is reduced 5–6 times [48].

The fructose:glucose ratio affects the speed and manner of honey crystallisation. In exemplary honeys from India studied by Naik et al., the ratio was 0.931, 1.17, 1.18, 1.23, 1.54. The most stable crystals were formed with the fructose:glucose ratio of 1.18, which was confirmed by simulations using artificial neural networks. Other ingredients that influence the manner of crystallisation are maltose, sucrose and water. This study proves that the recommended ratio (1.18) allows for a delay or even avoidance of the crystallisation process. It also enables us to understand the interactions of sugars present in honey using molecular dynamics [49].

- The paragraph 3.5. pH and free acidity should be implemented.

We expanded the paragraph with pH by adding information about the 2020 publication (verse 362-368)

With regard to pH and free acidity determination, no recent data on faster methods for analysing these parameters is available. As far as free acidity is concerned, newer techniques would make analysis easier as titration has to be performed quickly – within 2 minutes – which, in laboratory practice, is not always successful and requires the analysis to be repeated several times, which leads to the destruction of valuable samples. An innovative approach in the case of these parameters consists in data presentation. In 2020, Ratiu et al. published a study in which correlations between the analysed parameters, including pH, were shown using heat maps [54].

- A paragraph on generally bioactive compounds and antioxidant properties should be added.

We agree that it was worth adding information on other parameters determining antioxidant properties, so we added paragraph 3.9. (verses 445-460)

3.9. Other antioxidant properties and bioactive compounds

Both older and more recent publications are based on the colour assessment method in which the absorbance of a 50% honey solution is measured at a wavelength of 635 nm [66]. Additionally, the colour intensity of honey is assessed by measuring the color intensity ABS450 – the absorbance of a 50% honey solution at 450 and 720 nm [67]. Many authors correlate the obtained data with other parameters, looking for dependencies and explaining the biological activity of honey. Publications from recent years that evaluate the antioxidant properties of honey are based on previously developed methodologies and focus on, inter alia, determining the content of various compounds including flavonoids, ascorbic acid, carotenoids, β-carotene, lycopene, reducing sugar as well as assessing radical scavenging activity with 2,2-diphenyl-1-picrylhydrazyl (DPPH) [68], ferric-reducing/antioxidant power (FRAP) assay [69] and total antioxidant capacity [70]. These methods are based on spectrophotometric absorbance measurements. They are fairly fast and the obtained results are repeatable and therefore they constitute the basis of modern research on antioxidant properties. These properties are crucially important from the point of view of the prophylactic abilities of honey and their use in supporting the treatment of many diseases, which has been confirmed by numerous publications within the last 10 years [71, 72, 73].

Thanks again for any suggestions. We hope that the changes we have introduced will increase the substantive value of the work.

Yours faithfully,

Authors

Yours faithfully,
Authors

Round 2

Reviewer 2 Report

Major details should be added in Table 2

Paragraphes 3.8 and 3.9 should put together, reorganized and implemented including a Table

Author Response

We thank the reviewer for taking the time to read our work. We tried to take into account all suggestions. Below are the reviewer's recommendations and our responses.
- The type of paper should be changed from "Review" to " Perspective".
As suggested, we changed the manuscript type - verse 1
- The abstract should be better summarized.
To summarize the abstract better, we have added verses 25-28:
These methods are constantly modified, so that the honey that is on sale is a product of high quality. Prospects for devising methods of honey quality assessment include the development of a fast and accurate alternative to the melisopalinological method as well as quick tests to detect adulteration.
- The aim should explain the aim of this work and the novelty character.
To better define the aim, we have rewritten verses 63-66:
The aim of the present study was to summarise the most important methods of assessing honey quality published in the last 10 years. The novelty of these publication lies in the collection of data regarding both the evaluation of honey composition as well as adulteration.
- In the Conclusion limits, advantages , novelty and practical applications should be discussed.
To summarize better, we have added sentences about further perspectives and the need for research (verses 592-602),
Methods enabling the examination of several parameters which would prove the quality of bee honey within a short period of time are needed. Modern techniques which have been developed and validated over the past few years provide high precision and accuracy but need further improvement so that the natural bee honey available on the market can be tested as quickly and reliably as possible. Future research into methods of honey quality assessment should focus on developing techniques of rapid evaluation of the botanical origin of honey since the principal method currently in use, the melisopalinological method, is very time consuming, requires considerable experience, botanical knowledge as well as knowledge of the honey production process. It produces inconclusive results which can be difficult to interpret. Moreover, it is necessary to devise quick, reliable and inexpensive methods for detecting honey adulteration. As an overview of current methods of honey quality assessment, this paper can help shape the direction of future research in this field.
- Lines 31-39 should be enlarged.
In order to expand the paragraph better, we have added information on the types of honey and quality standards (verses 32-47)
Natural bee honey is a sweet product made by honeybees Apis mellifera L. both from the nectar of plants and from the excreta of insects sucking the juice from living parts of plants or from secretions of live parts of plants. Those are then combined with specific secretions of bees, stored in honeycombs, evaporated and left in honeycombs to mature. The Council of the European Union distinguishes nectar and honeydew honey as well as filtered honey, pressed honey, extracted honey, drained honey, comb honey and chunk honey or cut comb honey [1].
Bee honey intended for human consumption should be of sufficiently high quality. Requirements in individual countries include parameters such as variety determination, water content (no more than 20%; no more than 23% in heather and baker’s honey; no more than 25% in baker’s honey from heather), HMF (in principle, no more than 40 mg/kg), proline (no less than 25 mg/100 g), diastase activity (in principle, no less than 8 on the Schade scale), electrical conductivity (e.g. in honeydew honey no less than 0.8 mS/cm), pH, insoluble impurities (no more than 0.1 g/100 g) and free acidity (in principle, no more than 50 milli-equivalents acid/1000 g). Other parameters include, for example, colour and Total Phenolic Content (TPC). To assess honey quality, standard methods, including spectrophotometric, refractometric, titration and melisopalinological methods, are used [2].
- The paragraph 2. Honey composition is very poor! Example of previous work reporting honey composition and factors influencing honey quality should be inserted.
The composition we presented was very general, so we expanded it as suggested (verses 78-97).
Twenty five different sugars are found in honey. In nectar honeys, the sugars include erlose, maltose, sucrose and turanose, while in honeydew honey – melezitose and raffinose. Dextrins are found in Italian Metcalfa honey. Among the acids in honey, there is mainly gluconic acid, and also in smaller amounts: acetic, citric, formic, lactic, maleic, malic, oxalic, pyroglutamic and succinic acids. These acids influence the pH of honey, which is commonly between 3.3 and 4.6. Among the main minerals in honey, potassium should be mentioned (it constitutes approximately 33% of all elements present in honey). Others include barium (from 0.01 to 0.08 mg/100 g), boron (from 0.05 to 0.3 mg/100 g), chlorine (from 0.4 to 56 mg/100 g), cobalt (from 0.1 to 0.35 mg/100 g), fluoride (from 0.4 to 1.34 mg/100 g), iodine (from 10 to 100 mg/100 g), lithium (from 0.225 to 1.56 mg/100 g), molybdenum (up to 0.004 mg/100 g), nickel (up to 0.051 mg/100 g), rubidium (from 0.04 to 3.5 mg/100 g), silicium (from 0.05 to 24 mg/100 g), strontium (from 0.04 to 0.35 mg/100 g), sulfur (from 0.7 to 26 mg/100 g), vanadium (up to 0.013 mg/100 g) and zirconium (from 0.05 to 0.08 mg/100 g). The elemental composition is influenced by the origin – honeydew honeys are characterised by a higher content of elements. The most important and abundant amino acid in honey is proline. Moreover, enzymes such as amylase (diastase), invertase, catalase and glucose oxidase are present in honey. Enzymatic content proves the freshness of honey, and proper heating and storage conditions. Volatile compounds, both from the nectar and bee secretions, determine the smell of honey. Over 600 of them have been identified to date. By way of illustration, dark honeys contain more flavonoids. When
honey is stored at too high a temperature and in the process of prolonged heating (particularly honeys with a lower pH), the content of the unfavorable component, HMF, increases [20].
- Lines 127-137 should be clarified. data in Table 2 should be better discussed and major details should be added in table 2.
In order to better understand the methods we describe, we have developed the characteristics (verses 164-193). Additionally, in Table 3, we have added information on the botanical origin of the plants and the share of predominant pollen.
The MRJP2 gene has also been used to discriminate between two types of honey – honey produced by Apis mellifera and that produced by Apis cerana. The latter is far more expensive and therefore there have been cases of mislabelling honey in terms of origin. In order to identify them correctly, the authors designed two pairs of species-specific primers. The amplification products of A. mellifera and A. carena honeys were 560 and 212 bp, respectively. The obtained primers were characterised by high species specificity. The MRJP2 gene was detected using the PCR method and selected primers. Differences in the gene formed the basis for establishing the origin of honey. The PCR method enabled detection of the addition of A. mellifera honey which was as low as 1% [29].
Attempts have been made to distinguish between bee honey varieties using the CIE scale (Commission Internationale de l'Eclairage) L * C * abh ° ab [30]. The authors Tuberoso et al. assessed the colour of 17 monofloral honeys (n = 305). The advantage of the proposed method is small sample size (approximately 2 g) and absence of any destructive effect (samples can be reused in subsequent analyses). Principal Component Analysis (PCA) and hierarchical clustering analysis (HCA) have been used to distinguish between different honey varieties and classifications. In the case of reflection methods, the L * value for light honeys is below 50 while for dark honeys, it is above 50. The authors emphasise the fact that this technique may be particularly useful in the case of honeys with the protected designation of origin (PDO) mark. This concept covers honeys produced, processed and prepared in one area which have distinct characteristics from this area. Their names are legally protected and listed on the EU protected food name register [1]. The development of rapid tests which would assist in honey identification is particularly important since bee honey is one of the 25 products at highest risk of adulteration [31,32].
Another method of honey classification is fluorescence spectroscopy and statistical analyses based on parallel factor analysis (PARAFAC) and partial least squares method combined with discriminant analysis (DA PLS) [33]. It was used in a study by Lenhardt et al, which investigated acacia, sunflower, lime, meadows and artificial honey (n = 95). Fluorescence in the range of excitation (260– 290 nm) and emission (330–360 nm) comes from aromatic amino acids and is particularly important in differentiating the botanical origin of honey samples. The authors found that phenolic compounds and Maillard reaction products had the greatest impact on the distinction of individual varieties – emissions of these compounds differed most between honey varieties. Classification efficiency was approximately 90%. Additionally, artificial honey was detected 100% correctly.
- Lines 187-188 should be enlarged.
We expanded this paragraph as suggested (verses 250-255)
Other methods recommended for determining sugar content in bee honey include Gas Chromatography (GC) and HPLC with Pulsed Amperometric Detection. In the GC method, sugars are silylated and then the derivative fraction is quantified. Mannitol is used as an internal standard. The latter method (HPLC) is based on the principle that at high pH levels sugars behave like very weak acids – they are fully or partially ionised and therefore they can be separated using the ion exchange mechanism [2].
- Lines 237-243 should be better explained.
We have extended the description to better understand the methods under discussion (verses 203-316)
New methods of assessing moisture content in honey are based on evaluating the way of crystallisation by studying crystal size and shape. Tappi et al. utilised Differential Scanning Calorimetry (DSC) and time domain magnetic resonance (TD-NMR) methods. The TD-NMR method allowed the authors to distinguish two pools of protons, whose relative intensity was approximately 55% and 45%. It was also observed that static crystallisation is divided into two stages, with the second partially reversing the effects of the first. The study confirmed that in a dynamic process with continuous stirring, crystallisation time of honey is reduced 5–6 times [48].
The fructose:glucose ratio affects the speed and manner of honey crystallisation. In exemplary honeys from India studied by Naik et al., the ratio was 0.931, 1.17, 1.18, 1.23, 1.54. The most stable crystals were formed with the fructose:glucose ratio of 1.18, which was confirmed by simulations using artificial neural networks. Other ingredients that influence the manner of crystallisation are maltose, sucrose and water. This study proves that the recommended ratio (1.18) allows for a delay or even avoidance of the crystallisation process. It also enables us to understand the interactions of sugars present in honey using molecular dynamics [49].
- The paragraph 3.5. pH and free acidity should be implemented.
We expanded the paragraph with pH by adding information about the 2020 publication (verse 362-368)
With regard to pH and free acidity determination, no recent data on faster methods for analysing these parameters is available. As far as free acidity is concerned, newer techniques would make analysis easier as titration has to be performed quickly – within 2 minutes – which, in laboratory practice, is not always successful and requires the analysis to be repeated several times, which leads to the destruction of valuable samples. An innovative approach in the case of these parameters consists in data presentation. In 2020, Ratiu et al. published a study in which correlations between the analysed parameters, including pH, were shown using heat maps [54].
- A paragraph on generally bioactive compounds and antioxidant properties should be added.
We agree that it was worth adding information on other parameters determining antioxidant properties, so we added paragraph 3.9. (verses 445-460)
3.9. Other antioxidant properties and bioactive compounds
Both older and more recent publications are based on the colour assessment method in which the absorbance of a 50% honey solution is measured at a wavelength of 635 nm [66]. Additionally, the colour intensity of honey is assessed by measuring the color intensity ABS450 – the absorbance of a 50% honey solution at 450 and 720 nm [67]. Many authors correlate the obtained data with other parameters, looking for dependencies and explaining the biological activity of honey. Publications from recent years that evaluate the antioxidant properties of honey are based on previously developed methodologies and focus on, inter alia, determining the content of various compounds including flavonoids, ascorbic acid, carotenoids, β-carotene, lycopene, reducing sugar as well as assessing radical scavenging activity with 2,2-diphenyl-1-picrylhydrazyl (DPPH) [68], ferric-reducing/antioxidant power (FRAP) assay [69] and total antioxidant capacity [70]. These methods are based on spectrophotometric absorbance measurements. They are fairly fast and the obtained results are repeatable and therefore they constitute the basis of modern research on antioxidant properties. These properties are crucially important from the point of view of the prophylactic abilities of honey and their use in supporting the treatment of many diseases, which has been confirmed by numerous publications within the last 10 years [71, 72, 73].
Thanks again for any suggestions. We hope that the changes we have introduced will increase the substantive value of the work.
Yours faithfully,
Authors
